# Gradient and curl optical torques

Xiaohao Xu [1,2] ✉, Manuel Nieto-Vesperinas [3], Yuan Zhou [1,2], Yanan Zhang[1,2], Manman Li[1,2], Francisco J. Rodríguez-Fortuño [4,5], Shaohui Yan [1,2] ✉ & Baoli Yao [1,2] ✉

Optical forces and torques offer the route towards full degree-of-freedom manipulation of matter. Exploiting structured light has led to the discovery of gradient and curl forces, and nontrivial optomechanical manifestations, such as negative and lateral optical forces. Here, we uncover the existence of two fundamental torque components, which originate from the reactive helicity gradient and momentum curl of light, and which represent the rotational analogues to the gradient and curl forces, respectively. Based on the two components, we introduce and demonstrate the concept of lateral optical torques, which act transversely to the spin of illumination. The orbital angular momentum of vortex beams is shown to couple to the curl torque, promising a path to extreme torque enhancement or achieving negative optical torques. These results highlight the intersection between the areas of structured light, Mie-tronics and rotational optomechanics, even inspiring new paths of manipulation in acoustics and hydrodynamics.

Accurate control over the dynamics of matter has been the goal of scientists involved in the development of optical manipulation techniques[1–3]. In the last decades, the field continually advanced as the understanding of several optical force components progressed, as well as, to some extent, concerning the optical torque. The simplest and long ago well-known illustration is the radiation pressure exerted by structureless light on a particle, pushing it along the wave propagation direction (cf. Figure 1a). Similarly, the action of a torque on the particle is known when the incident wave possesses a spin angular momentum (SAM), (which we shall indistinctly refer to just as spin), namely, when its polarization rotates around its direction of propagation. In both dynamical effects, the canonical example is the plane wave, which, as such, imposes a fundamental limit on the force and torque, including their direction and magnitude[4].

Structured illumination enriches these mechanical effects, there being much more room for force orienting and enhancement. For example, the gradient force,arising on illumination with focused light (Fig. 1b), can be enhanced by control of the intensity inhomogeneity to overcome the radiation pressure;this ledto the advent of optical

tweezers[5,6]. It is known that both the electric and magnetic fields can contribute to this force[7–9].

Also, for particles with interacting induced multipoles, it was found in nonparaxial beams that the radiation pressure may become a negative -i.e., pulling- optical force (NOF), which allows for a long-range particle delivery towards the light source[10–12]. When the field possesses a spin-curl, or Belinfante spin momentum, the curl force will be available[7,13–15], and with evanescent waves[16] it can transport the particle transversely to the propagation direction[16–18], a phenomenon known as lateral optical force (LOF)[19–22]. The LOF can also be realized by utilizing the spin part of the imaginary Poynting momentum[7,16,23], which is ubiquitous in structured fields[8,17,18,24–26].

While the optical force governs the center-of-mass motion of the particles, the (intrinsic) optical torque enables control over rotation with respect to their own axis. In contrast to the force, however, excepting the optical SAM, there seem to be very few field properties that can be harnessed for the torque, especially for small particles[27–31]. This has forced researchers to exploit complex sets of particles or unusual (e.g. anisotropic or chiral) materials to achieve a negative optical torque (NOT)[32–39],which rotates the object contrary to the SAM

[1]State Key Laboratory of Transient Optics and Photonics, Xi'an Institute of Optics and Precision Mechanics, Chinese Academy of Sciences, Xi'an 710119, China. [2]University of Chinese Academy of Sciences, Beijing 100049, China. [3]Instituto de Ciencia de Materiales de Madrid, Consejo Superior de Investigaciones Científicas, Campus de Cantoblanco, Madrid 28049, Spain. [4]Department of Physics, King's College London, Strand, London WC2R 2LS, UK. [5]London Centre for Nanotechnology, Department of Physics, King's College London, Strand, London WC2R 2LS, UK. ✉e-mail: xuxiaohao@opt.ac.cn; shaohuiyan@opt.ac.cn; yaobl@opt.ac.cn

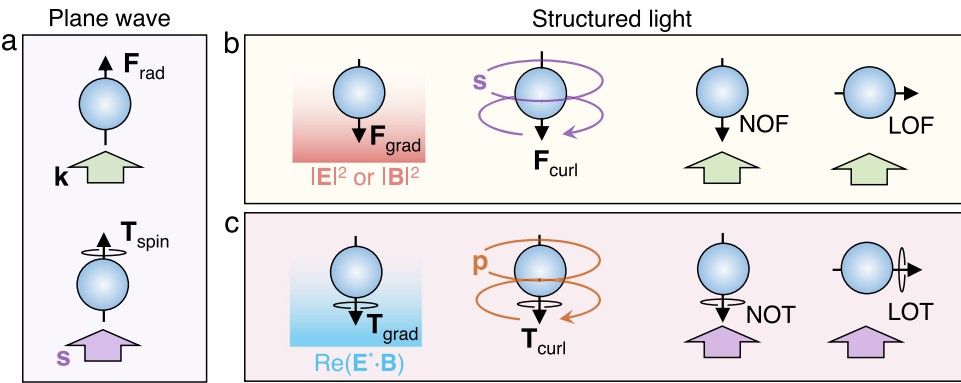

**Fig. 1 | Mechanical effects of light on small particles. a** For plane wave illumination, the optical force $\mathbf{F}_{\text{rad}}$ is provided only by radiation pressure which acts in the propagation direction, and the optical spin dictates the torque $\mathbf{T}_{\text{spin}}$. **b** Under structured light, gradient and curl forces may arise due to the field intensity gradient and spin-curl, it being also possible to realize the concepts of negative (NOF) and lateral (LOF) optical force, characterized by force directions opposite and orthogonal to the propagation, respectively. **c** The present work puts forward the rotational analogues of the forces shown in (**b**). We discover the gradient and curl torques, stemming from the inhomogeneity of the reactive helicity and electromagnetic (i.e., Poynting) momentum, respectively; showing how to harness them to create the negative (NOT) and the lateral (LOT) optical torque.

of the illuminating wavefield and, hence, constitutes a rotational analog to the NOF; but the angular counterpart of the LOF has still been elusive.

In this work, by establishing an incident field-featured model of optical torque, we reveal two fundamentally new components (cf. Figure 1c), which are derived from the curl of the electromagnetic (i.e. Poynting) momentum and the gradient of the reactive helicity[16], which interestingly are the angular analogues of the spin-curl and the well-known gradient force, respectively. We propose two structured light fields to respectively observe these gradient and curl components, and show that it is possible to generate a net torque that acts transversely to the incident SAM. This transverse net torque, which we refer to as lateral optical torque (LOT), represents the angular analog of the LOF. In addition, we show that the curl torque in vortex beams is subject to the orbital angular momentum (OAM), even though the particle is small and placed off the beam axis. This is in marked contrast to the generalized belief that the OAM is only responsible for the azimuthal force on off-axis trapped particles[27]. The curl torque as such can be enhanced by increasing the topological charge, irrespective of the local intensity. It may oppose the spin-induced component, and even overcome it, resulting in a NOT that rotates the object against the incident spin. Our findings are applicable to simple Mie particles and are verified by numerical computations.

## Results

### Spin-independent torques on multipoles

Throughout this paper, we base our results on the interaction of an isotropic neutral sphere and monochromatic waves (with angular frequency $\omega$ and wave number $k = \omega/c$). We are interested in the time-averaged intrinsic optical torque, which rotates particles about their own axis. For a nonchiral dipolar particle embedded in a medium with permittivity $\varepsilon$ and unity permeability, the torque is known to be[34,40–42]:

$$\mathbf{T} = \frac{c}{2}\left[ C_{\text{abs}-\text{e}}^{(1)}\mathbf{s}_{\text{e}} + C_{\text{abs}-\text{m}}^{(1)}\mathbf{s}_{\text{m}} \right], \qquad (1)$$

where $C_{\text{abs}-\text{e}}^{(1)}$ and $C_{\text{abs}-\text{m}}^{(1)}$ are the absorption cross-sections due to electric and magnetic dipoles induced in the sphere, while $\mathbf{s}_{\text{e}} = \varepsilon(\mathbf{E}^* \times \mathbf{E})/(i\omega)$ and $\mathbf{s}_{\text{m}} = \varepsilon c^2(\mathbf{B}^* \times \mathbf{B})/(i\omega)$ are the incident electric and

magnetic spin at the particle center. Equation (1) indicates that the torque on a dipolar sphere is determined solely by the optical spin. Since the absorption cross-sections are always positive scalars, there is no NOT nor LOT on a dipolar particle.

We then consider a general case, in which the torque is written as the sum of the contribution from all multipoles induced in the sphere[42,43]: $\mathbf{T} = \sum_{l=1}^{N}\mathbf{T}^{(l)}$, where $\mathbf{T}^{(l)}$ is the component due to electric and magnetic $2^l$-poles, and $N$ is the multipole series truncation index, i.e., the maximum order of multipoles included. In Lorenz-Mie theory[43], $\mathbf{T}^{(l)}$ is given by the expansion coefficients of the incident and scattered fields, while the Cartesian multipole expansion method expresses the torque by the multipole moments and incident field vectors[42]. However, both theories do not showcase an explicit relation between the torque and the incident field properties (e.g., the spin).

Using the angular spectrum method, herein we provide an insightful model, which classifies the torque into three categories (see Supplementary Note 1 for details):

$$\begin{aligned}
\mathbf{T} &= \mathbf{T}_{\text{spin}} + \mathbf{T}_{\text{grad}} + \mathbf{T}_{\text{curl}} \\
&= \sum_{l=1}^{N} \mathbf{T}_{\text{spin}}^{(l)} + \mathbf{T}_{\text{grad}}^{(l)} + \mathbf{T}_{\text{curl}}^{(l)}.
\end{aligned} \qquad (2)$$

where

$$\begin{aligned}
\mathbf{T}_{\text{spin}}^{(l)} &= \hat{\mathcal{A}}_{\text{e}}^{(l)}\mathbf{s}_{\text{e}} + \hat{\mathcal{A}}_{\text{m}}^{(l)}\mathbf{s}_{\text{m}}, \\
\mathbf{T}_{\text{grad}}^{(l)} &= \hat{\mathcal{A}}_{\text{grad}}^{(l)}\nabla\mathcal{H}, \\
\mathbf{T}_{\text{curl}}^{(l)} &= \hat{\mathcal{A}}_{\text{curl}}^{(l)}\nabla \times \mathbf{p},
\end{aligned} \qquad (3)$$

with $\mathbf{p} = \varepsilon\mathcal{R}(\mathbf{E}^* \times \mathbf{B})$ and $\mathcal{H} = \varepsilon\Re(\mathbf{E}^* \cdot \mathbf{B})/k$ being the density of electromagnetic momentum and the reactive helicity[16] (tentatively called magnetoelectric energy in[44]) of the incident wave, respectively. The prefactors are operators described by the Laplacian $\Delta$ and the particle constitutive parameters, (cf. Eq. (33) in Supplementary Note 1).

For $l = 1$ (dipoles), $\hat{\mathcal{A}}_{\text{grad}}^{(l)} = \hat{\mathcal{A}}_{\text{curl}}^{(l)} = 0$, and $\mathbf{T}_{\text{spin}}^{(l)}$ reduces to Eq. (1). Hence, in contrast with the optical force[7,13,22], there is no gradient nor curl torque acting on a dipolar particle. On the other hand, for

quadrupoles ($l = 2$), they are given by

$$\hat{A}_e^{(2)} = c\left[C_{abs-e}^{(2)}(1+\frac{\Delta}{k^2}) - \frac{1}{2}C_{abs-m}^{(2)}\right],$$

$$\hat{A}_m^{(2)} = c\left[C_{abs-m}^{(2)}(1+\frac{\Delta}{k^2}) - \frac{1}{2}C_{abs-e}^{(2)}\right],$$

$$\hat{A}_{grad}^{(2)} = \frac{c}{k}\left[C_{abs-e}^{(2)} - C_{abs-m}^{(2)}\right],$$

$$\hat{A}_{curl}^{(2)} = \frac{c}{k^2}\left[C_{abs-e}^{(2)} + C_{abs-m}^{(2)}\right]. \tag{4}$$

where $C_{abs-e}^{(2)} = \frac{10\pi}{k^2}[\Re(a_2) - |a_2|^2]$ and $C_{abs-m}^{(2)} = \frac{10\pi}{k^2}[\Re(b_2) - |b_2|^2]$, with $a_2$ and $b_2$ being the second order Mie coefficients, are the absorption cross-sections due to electric and magnetic quadrupoles.

Equations (2) and (3) reveal how the structure of illumination determines the torque. We see from $\mathbf{T}_{spin}^{(l)}$ in Eqs. (3) and (4) that for high-order ($l > 1$) multipoles, the spin-induced torque generally exists and either the electric or magnetic spin gives rise to it via both $C_{abs-e}^{(l)}$ and $C_{abs-m}^{(l)}$ [cf. Eqs. (3) for $\mathbf{T}_{spin}^{(l)}$ and (4)]. This differs from the case of dipoles, Eq. (1), in which the electric (or magnetic) spin couples to the torque by only the respective electric (or magnetic) absorption cross-section. Also, the Laplacian $\Delta$ in Eqs. (4) indicates that the torque will be affected by the spin in the vicinity of the particle center. Such a nonlocal effect is also seen in the multipolar optical force[8,15].

Most importantly, Eqs.(3) uncover the existence of gradient and curl torques, $\mathbf{T}_{grad}^{(l)}$ and $\mathbf{T}_{curl}^{(l)}$, which originate from the gradient of the reactive helicity $\nabla\mathcal{H}$ and the curl of electromagnetic momentum $\nabla \times \mathbf{p}$, respectively; this is in intriguing analogy with the intensity gradient and spin-curl forces. It is also remarkable that the rotational analogue of the intensity gradient force is the gradient of a reactive quantity recently discovered[16].Both torques are in general spin-independent; however, their generation entails the excitation of multipoles higher than the dipoles, because their prefactors, $\hat{A}_{grad}^{(l)}$ and $\hat{A}_{curl}^{(l)}$, are zero for $l = 1$ [see Eq. (33) in Supplementary Note 1 for details]. In this connection, it is worth noting that the gradient and curl terms appear in the dipolar contributions from the conservation of SAM and of OAM, separately, but they are always cancelled out in the net torque when the total angular momentum conservation is fully considered[40,41], as also indicated by Eq. (1). Also, recent progress in field theory has established the momentum curl as the 'Poynting' part of the dual-symmetric spin[45,46], and it is known that the spin can be proportional to the momentum curl for some special fields (e.g., TE and TM evanescent waves)[47,48].

## Lateral optical torque (LOT)

A direct observation of the gradient and curl torques requires structured light illumination. To this end, we consider an incident wavefield of electric vector:

$$\mathbf{E} = \frac{E_0}{k}\left[k_z u(x,y)\mathbf{e}_x + i\partial_x u(x,y)\mathbf{e}_z\right]e^{ik_z z}, \tag{5}$$

where $E_0$ denotes the real amplitude, $(K, k_z) = k(\sin\theta, \cos\theta)$, (cf. Figure 2a), and $u(x,y) = \cos(Kx) + i\sin(Ky)$. The magnetic field is readily found to be:

$$\mathbf{B} = \frac{E_0}{ck^2}\Big\{ \left[k^2\cos(Kx) + ik_z^2\sin(Ky)\right]\mathbf{e}_y$$
$$- Kk_z\cos(Ky)\mathbf{e}_z\Big\}e^{ik_z z}. \tag{6}$$

In practice, the field can be constructed by the interference of four plane waves (consisting of two $s$-polarization and two $p$-polarization waves): $[\mathbf{E}_1, \mathbf{E}_2, \mathbf{E}_3, \mathbf{E}_4] = \frac{E_0}{2k}[k_z\mathbf{e}_x e^{iKy}, -k_z\mathbf{e}_x e^{-iKy}, (k_z\mathbf{e}_x - K\mathbf{e}_z)e^{iKx}, (k_z\mathbf{e}_x + K\mathbf{e}_z)e^{-iKx}]e^{ik_z z}$, with their wave vectors distributed on a cone of half-cone angle $\theta$ as illustrated in Fig. 2a.

It can be verified that this field has vanishing $z$-component spin over the whole space: $s_{e,z} = s_{m,z} = 0$. However, the properties of the momentum-curl and reactive helicity gradient highly depend on the value of the cone angle $\theta$ (or $k_z$). For example, the $z$-component momentum curl is proportional to $(k_z^2 + |k_z|^2)\sin(Kx)\cos(Ky)$, which is zero when $k_z$ is imaginary but nonzero for real $k_z$. We thus distinguish two scenarios:

Case I, the interference yields a traveling wave, (real $\theta : 0 < \theta < \pi/2$). Then we have $\nabla\mathcal{H} = 0$, and

$$(\nabla \times \mathbf{p})_z \propto \sin(Kx)\cos(Ky), \tag{7}$$

which is plotted in the color map in Fig. 2a.

Case II, the interference yields a standing evanescent wave, (complex $\theta : k_z = k\cos\theta = iq$), where $q$ is the decay constant along the positive $z$-axis[16]. Then the $z$-component of the reactive helicity gradient appears,

$$(\nabla\mathcal{H})_z \propto \sin(Kx)\cos(Ky)e^{-2qz}, \tag{8}$$

which exhibits the same $x$- and $y$-dependent properties as $(\nabla\times\mathbf{p})_z$ in case I, but the momentum curl now vanishes in the $z$ direction: $(\nabla\times\mathbf{p})_z = 0$. Note that there is no propagation for this case, because the propagation vectors of the four waves cancel each other.

Furthermore, Fig. 2b, c illustrate the spatial distribution of all these quantities of interest for both the traveling wave ($\theta = \pi/4$) and the evanescent wavefield ($\theta = \arccos(0.4i)$). Evidently, in either case, a $z$-component torque $T_z$, will be produced, as inferred from Eq. (3). For case I, $T_z$ is produced from the momentum curl $\nabla \times \mathbf{p}$ because only this quantity possesses the component in the $z$-direction. Therefore, the traveling wave offers an ideal platform for the observation of the curl torque. Likewise, the evanescent wave (case II), in which only the reactive helicity gradient $\nabla\mathcal{H}$ has the vertical component, allows us to directly probe the gradient torque.

It is remarkable that in either case $T_z$ is transverse to the spin of illumination. This reminds us of the force counterpart: the LOF, which acts in the direction where the illumination has no propagation, and which is now a subject of active research in optical manipulation[19–22,25,49,50]. For this reason, we coin $T_z$ the lateral optical torque (LOT). We emphasize that the LOT defines the torque orthogonal to the spin, rather than to the propagation direction.

Figure 2d shows the three coordinate components of the total torque $\mathbf{T}$, which is calculated with the prevailing Lorenz-Mie method, for a magnetoelectric Si nanoparticle probe (radius $r = 0.1\,\mu m$). Because of its low absorption in the near-infrared region, we have employed a visible excitation wavelength ($\lambda = 0.58\,\mu m$) for this material. The truncation index is set as $N = 10$, which suffices for an exact computation for this particle, and the field amplitude is assumed to be $E_0 = 10^7$ V/m. As expected, in both cases the LOT (or $T_z$) is nonzero in general, and it is maximized where both $T_x$ and $T_y$ vanish. This is in favor of an experimental observation of the two torques. We stress that in either case the LOT represents the lateral component of the net torque, because it is derived from Supplementary Eq. (2), which is a result of the conservation of total optical angular momentum.

To explore the multipole resonance effects, we calculated the LOT with varying particle sizes and wavelengths. The particle is placed at the position of the torque maximum. The results are shown in Fig. 2e,f for traveling (case I) and evanescent (case II) excitation, respectively. For both cases, multiple magnitude maxima appear due to the rich number of Mie resonances[51] supported by high-index dielectric particles. However, the LOT is zero in the region of long wavelengths and small radii, where the dipole modes dominate (see Supplementary Fig. S1 for the absorption spectrum), because the dipole term produces no LOT. The lowest-order multipoles required for the LOT are identified as the electric (EQ) and magnetic (MQ) quadrupole. This is well understood

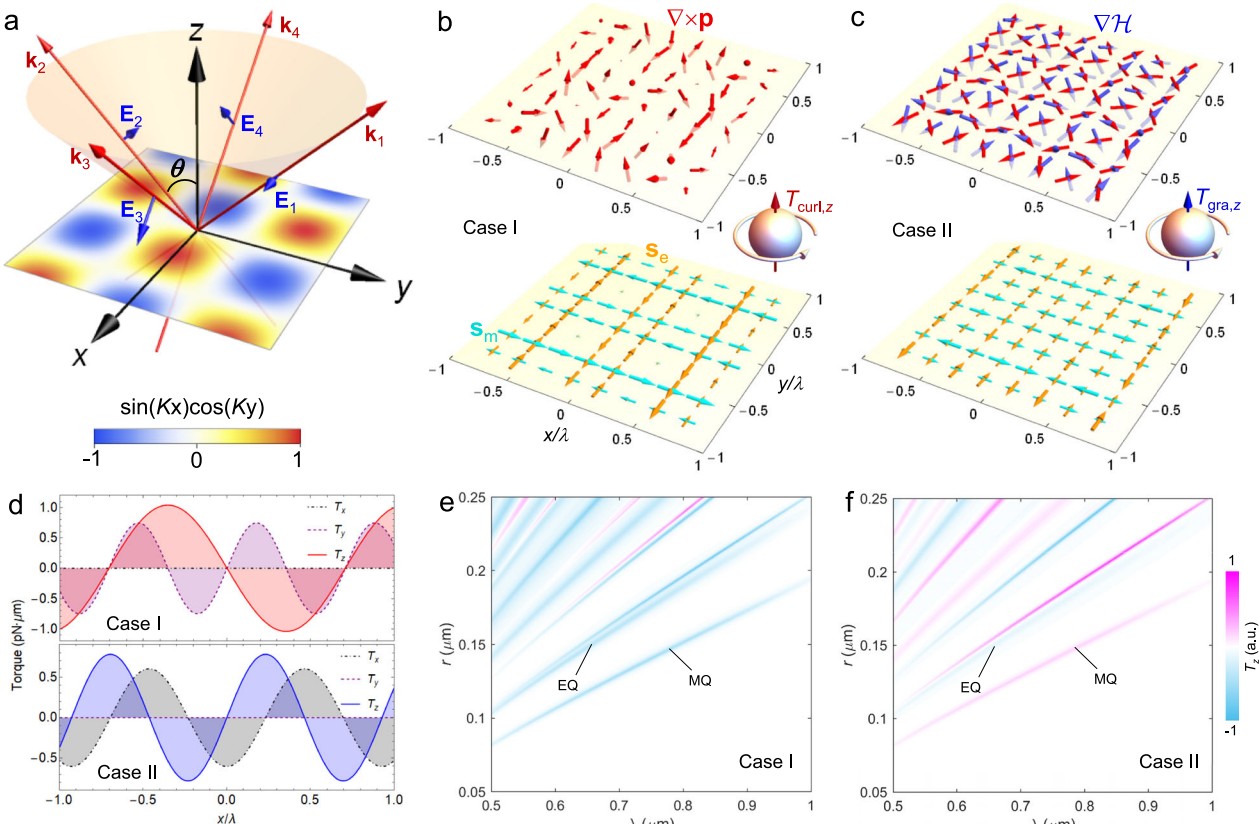

**Fig. 2 | Manifestation of gradient and curl torques as the LOT. a** Incident structured light field consisting of two pairs of $s$- and $p-$ polarized waves. These four waves share an equal amplitude, wave number, and incidence angle $\theta$ with respect to the $x-y$ plane. The color scale plot represents the $z-$ component of the momentum-curl $\nabla \times \mathbf{p}$ for the traveling wave (case I), as well as of the reactive helicity gradient $\nabla \mathcal{H}$ for the standing evanescent wave(case II). **b, c** Field quantities of interest for the traveling (**b**) and evanescent (**c**) waves. Insets illustrate the expected LOT due to $\nabla \times \mathbf{p}$ in (**b**), and $\nabla \mathcal{H}$ in (**c**). **d** Calculated Cartesian components of the total torque for $\lambda = 0.58\,\mu m$ on a Si particle (radius $r = 0.1\,\mu m$) placed on the $x$-axis. **e, f** The LOT as a function of the wavelength and particle radius for cases I (**e**) and II (**f**). The torques are normalized to their maxima (7.0 and 14.6 pN $\cdot\,\mu m$ for I and II, respectively).

because of the high-order multipole nature of both the curl and gradient torques, [see Eq. (3)], and in fact we did not observe the LOT when taking $N = 1$ in the Lorenz-Mie calculations, as it should. In addition, the sign of the LOT is switchable by tuning the resonance conditions. Interestingly, the torques in the EQ and MQ channels have the same sign in case I, but in case II the torque sign in MQ is opposite to that in EQ. This is explained by the dual-symmetric and dual-antisymmetric properties of the prefactors, $\hat{\mathcal{A}}_{\text{curl}}^{(l)}$ and $\hat{\mathcal{A}}_{\text{grad}}^{(l)}$, for the curl and gradient torque [see Eq. (4)]. Ultimately, these results based on the Lorenz-Mie method can be reproduced using our analytical model [Eq. (3)].

## LOT on anisotropic particles

The above sphere-based LOT will not depend on the particle orientation because of isotropy. Realistic particles cannot be perfect spheres and rod-like geometries (e.g., dimers and cylinders) thus being of interest in experiments[28,52,53]. For anisotropic particles, the torque should be sensitive to the orientation and torsional balance (or restoring effect) might occur[52].

To simulate realistic spherical particles, we consider spheres of varying roughness (see particles i-iv in Fig. 3a), with the illumination of traveling wave ($\lambda = 0.58\,\mu m$) as an example. The optical torque is computed based on the Maxwell stress tensor method[23], in which the electromagnetic field is obtained by finite-difference time-domain (FDTD) simulations. Figure 3b shows the calculated LOT on the particles with varying orientation angle $\alpha$ at $y = 0$ and $x = 0.35\,\mu m$, where the momentum curl of illumination is maximized. The $\alpha$-independent torque on the isotropic particle i is approximately $-1.0$ pN $\cdot\,\mu m$, in good

agreement with our analytical theory (cf. red line in Fig. 2d). Now, due to the anisotropy, the torque magnitude varies with $\alpha$ on the rough spheres ii-iv, and larger roughness yields more significant variation. However, the torque maintains its negative sign, irrespective to the orientation. This indicates that the rough spheres will exhibit the running state to rotate clockwise and continuously, consistent with the perfect sphere, as shown by the angular potential (Fig. 3c), which is calculated by the opposite angular integral of the torque. It also suggests that the restoring contribution of the torque is overwhelmed by that responsible for continuous rotation. Therefore, our theory can well predict and explain the dynamic behaviors of these sphere-like structures.

We then evaluate the LOT and angular potential for the dimers (i.e., particles v and vi in Fig. 3a) made of two identical Si spheres of radius $r$. For this highly anisotropic geometry, it is instructive to extract the nonconservative and conservative (or restoring) parts from the torque (see Methods for details). The results for the smaller dimer (particle v) are shown in Fig. 3d. The total LOT, $T_z$, varies with the orientation angle in a sinusoidal-like way, and it changes sign with a negative derivative around $\alpha = 90°$ or $270°$. Accordingly, an angular potential well is formed, by which the particle exhibits the locked (torsional) state (Fig. 3f). It is also noted that the washboard potential is slightly tilted due to the presence of a small nonconservative component $T_z^{\text{ncons}}$ in Fig. 3d. However, the dynamic behavior of the larger dimer (particle vi) is quite different. As shown in Fig. 3e, $T_z$ is always negative, because the magnitude of $T_z^{\text{ncons}}$ dominates over that of $T_z^{\text{cons}}$. As a result, the potential well vanishes, switching the dimer to the running state (Fig. 3f).

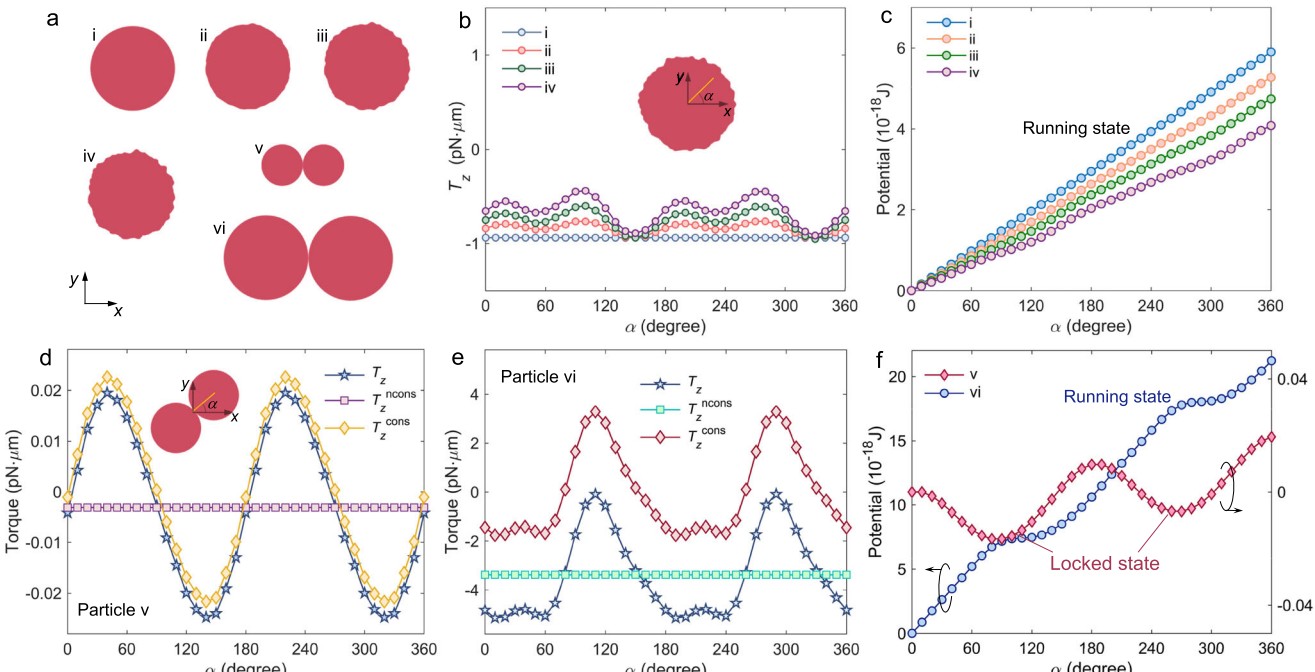

**Fig. 3 | FDTD simulation results for different shaped particles. a** Sectional view of particles used in the simulations. Particle i represents an ideal sphere with radius of 0.1 $\mu m$; particles ii-iv are rough spheres modeled by varying radius values, which converge in distribution to a Gaussian variable with mean 0.1 $\mu m$ and standard deviation $\sigma = 1$, 2 and 3 nm, respectively; particles v and vi are dimers composed of dual spheres with radius of 0.05 and 0.1 $\mu m$, respectively. **b** LOT as a function of the orientation angle $\alpha$ of the spherical particles i-iv. The angle is defined with respect to *x*-axis, as illustrated by the inset. **c** Calculated angular potential for the particles i-iv. **d** LOT and its conservative and nonconservative components versus the orientation angle of the dimer v. **e** A similar plot for the dimer vi. **f** Angular potential for v and vi. In all simulations, the particles are illuminated by the travelling wave (case I) in Fig. 2a, and are placed at a fixed position $x = 0.35\lambda$, where the momentum curl of illumination reaches the maximum.

Conventionally, the running state and the washboard potential tilt are attributed to the circular polarization (or optical spin) of light[52]. They are, however, explained by the momentum curl in our case, because only this quantity of illumination has the lateral component (see Supplementary Note 2 for detailed proof based on *T* matrix theory[54,55]). It also interprets the dominant restoring effect on the small dimer, because the momentum curl produces torque through higher multipole responses, which are weak for reduced particle size.

### Torque in optical spanner

The existence of the gradient and curl torques motivate us to reexamine the principle of the optical spanner, which is an optical manipulation tool for particle rotation. It has been well known that small particles trapped off-axis in a light beam carrying both SAM and OAM, can orbit about this axis and simultaneously rotate with respect to their own axis. There is a generalized belief that the orbiting and self-rotation are caused by the local OAM and SAM, respectively[27,56,57]. This matches our intuition since the local OAM density is extrinsic while the spin is intrinsic. Despite the observation that the torque on large particles trapped on-axis (or at the vortex center) is sensitive to the OAM[53,58,59], it can be explained that once the particle is located at the vortex center of light impinging on it, the total OAM becomes intrinsic due to geometrical symmetry[60]. However, as we will show below, actually the global OAM may still contribute to the torque even when a small particle is placed off the vortex center.

To illustrate this, let us consider a perfect vortex beam obtained by focusing the field at the pupil plane:

$$\mathbf{E}_{input} = E_0 J_\ell(k_\varrho \rho_0/f) e^{i\ell\varphi} \mathbf{u}, \qquad (9)$$

with focal length *f*, topological charge $\ell$ and $\rho_0$ determining the radius of the ring-like focused field at the focal plane; $(\varrho, \varphi)$ are the polar coordinates of the objective lens' input pupil, and **u** and $J_\ell(\cdot)$ denote the state of polarization and the first kind of $\ell$-order Bessel function, respectively. Analytically, one may write the field in the focal plane, by neglecting the other two components orthogonal to **u** (which are small), as: $\mathbf{E} = h(\rho, \rho_0) e^{i\ell\phi} \mathbf{u}$, where $(\rho, \phi)$ are the cylindrical coordinates. This perfect vortex field is characterized by a vortex phase $e^{i\ell\phi}$ and an annular intensity profile $I = |h(\rho, \rho_0)|^2$ with a fixed radius $\rho_0$ (independent of $\ell$)[15]. It is evident that for linear polarization $\mathbf{u} = \mathbf{e}_x$, there will be no spin. However, due to the helical phase or OAM, the kinetic momentum is able to acquire an azimuthal component $p_\phi$, spiraling around the beam axis[27]. Such circulation will give rise to a nonzero axial momentum-curl, as

$$(\nabla \times \mathbf{p})_z \propto \frac{1}{\rho}\frac{\partial}{\partial\rho}(\rho p_\phi) \propto \ell \frac{\partial I}{\rho \partial \rho}. \qquad (10)$$

It follows directly from the curl component in Eqs. (3) that the axial intrinsic torque can still be produced even in the absence of spin.

To see whether the curl torque is significant in the optical spanner, it is important to compare this nontrivial component with the conventional spin-induced torque. To this end, we then consider left-handed circular polarization, $\mathbf{u} = \sqrt{2}(\mathbf{e}_x + i\mathbf{e}_y)/2$ in Eq. (9), so that the spin coexists with the momentum-curl. Figure 4a shows the field distribution at the focal plane, for this vortex beam ($\ell = 10$, $\lambda = 1.064 \mu m$) focused with a numerical aperture NA = 0.95, which is calculated rigorously by the Richards-Wolf method[8,15]. The field intensity has peaked at a predesigned radius $\rho_0 = 4.0 \mu m$, with vortex-like Poynting momentum. The electric and magnetic spins take a positive value, and in fact they are equal everywhere for both their signs and magnitudes. However, the momentum-curl component, Eq. (10), may change its sign as this quantity is associated with the intensity inhomogeneity.

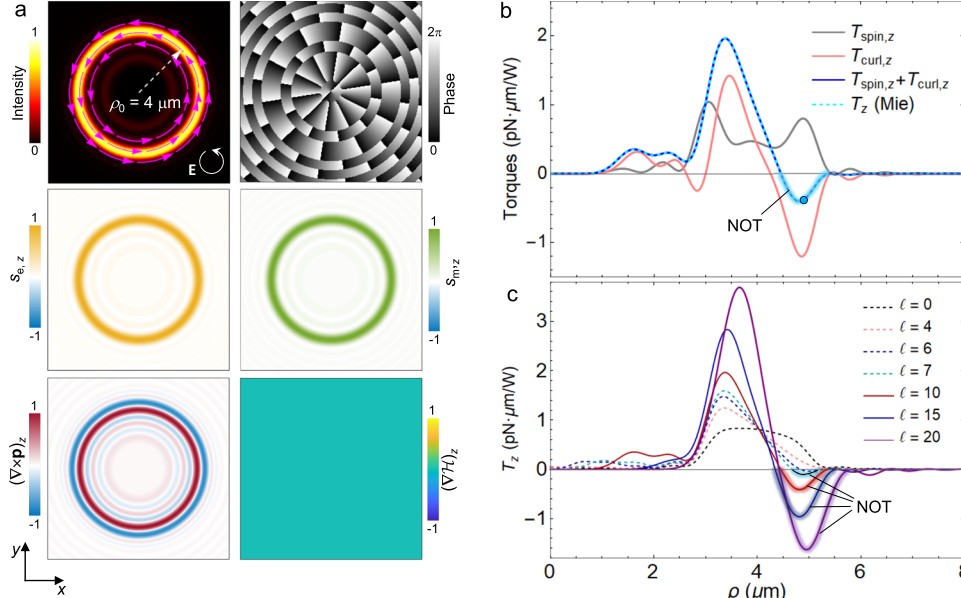

**Fig. 4 | NOT on homogeneous isotropic spheres in perfect vortex beams.**
**a** Calculated field distribution on the focal plane of a perfect vortex beam
($\lambda = 1.064\,\mu$m) with left-handed circular polarization and a topological charge $\ell = 10$. The field quantities are normalized to the maximum of their absolute values. Magenta arrows in the intensity map indicate the direction of transverse momentum density. **b** Calculated longitudinal SAM-induced and momentum-curl torques

on a gold particle ($r = 0.75\,\mu$m) placed at varying radial positions. Their sum is also presented, which agrees with Lorenz-Mie results for the net torque $T_z$. NOT occurs when $T_z$ is negative, i.e., against the incident angular momentum. The blue circle indicates the radial trapping position. **c** Calculated net longitudinal torques for different topological charges.

Note that the reactive helicity gradient vanishes for this field at the focal plane.

Figure 4 b shows the longitudinal total torque $T_z$ and its spin and curl components, $T_{\text{spin},z}$ and $T_{\text{curl},z}$, on a gold sphere of radius $r = 0.75\,\mu$m. The sum of $T_{\text{spin},z}$ and $T_{\text{curl},z}$, which are calculated with Eqs. (2) and (3), matches exactly with $T_z$ obtained by the Lorenz-Mie method, validating our theory. The spin component, as expected, is always positive for left-handed circular polarization, following the direction of the SAM. However, $T_{\text{curl},z}$ can act against $T_{\text{spin},z}$, and dominate over the latter. Consequently, the NOT is achieved at $4.5 < \rho < 5.5\,\mu$m. This NOT is opposite not only to the spin, but also to the total angular momentum of the illumination wave, because both the SAM and OAM are positive in our configuration. We note that the NOT region includes the position of radial optical trapping ($\rho = 4.8\,\mu$m), which is obtained by evaluating the radial force (see Supplementary Fig. S2 for details). Multipolar analysis of the curl torque by Eq. (3) indicates that the NOT at the trapping position is mainly attributed to the 16- and 32-multipoles (cf. Supplementary Fig. S3). We remark that lower-multipole excitation (e.g., using smaller plasmonic nanoparticles at visible illumination) may also generate the NOT, but requires larger topological charge $\ell$.

In fact, the NOT can be enhanced by increasing $\ell$ of the incident OAM, and be canceled by decreasing it to $\ell < 6$ so that the torque becomes positive everywhere, (Fig. 4c). This is qualitatively explained by Eq. (10), by which the momentum-curl (and therefore the curl torque) increases with $\ell$. Finally, it is found that when the circular polarization is switched to right-handed, the NOT is equally achievable, as shown in Supplementary Fig. S4.

## Discussion

We have built a multipole theory for the optical torque, with a classifying framework featuring three fundamental field properties: the optical spin, the reactive helicity gradient, and the Poynting-momentum curl. We affirm them as "fundamental" for the following reasons. First, these three aforementiond quantities are linearly independent, so that each of them cannot be constructed by the others.

Second, they stem from well-established concepts extensively employed in the study of field theory[16,44–47]. Most importantly, they form a set that constitutes a complete vector basis of the torque on one of the most fundamental particles: a sphere.

On these grounds, our theory establishes two fundamental dynamic quantities: the curl and the gradient torque, whose strength may even overcome the spin-imposed constraint on particle rotation. To directly detect them, we have proposed two structured light fields, from which these two torques act in the direction transverse to the SAM, a phenomenon that we termed lateral optical torque (LOT), which is a rotational counterpart of the well-known lateral optical force (LOF)[19–22,25,49,50,61]. Although we have uncovered the curl and gradient components using a particle model of the isotropic sphere, we have shown that they can also be exerted on nonspherical structures, contributing to the nonconservative torque (see Fig. 3 and Supplementary Note 2). This makes them useful in explaining or predicting the rotational behaviors of anisotropic particles in light.

With these two quantities, we address the principle of optical spanners[27,56,62,63], showing how the optical OAM can couple to the curl torque, counterintuitively spinning objects placed off-axis. Furthermore, we have demonstrated the competition between the curl torque and the spin-induced torque in a beam carrying both OAM and SAM. By controlling the topological charge, the curl torque is shown to overwhelm the spin one, resulting in a NOT whose direction is opposite to the incident spin or total angular momentum. This NOT is achieved for single normal spheres, in contrast to previous schemes based on speciallydesigned (anisotropic or core-shell) objects or particle arrays[32,33,35–39].

Compared with the conventional spin-induced torque, whose magnitude is limited by light intensity[4] due to the universal inequality: $|\mathbf{E}^* \times \mathbf{E}| \leq |\mathbf{E}|^2$, the gradient and curl torques circumvent this limitation because they can be enhanced by taking advantage of the field inhomogeneity, in analogy to the gradient[5] and curl forces[7,13,17,18]. Particularly, the magnitude of the curl torque can even, in principle, potentially grow without limit in the OAM beam, by increasing the topological charge or the phase inhomogeneity along the azimuthal direction. It is

plausible that reactive fields, i.e., those mainly containing evanescent components like near-fields or surface waves, yield larger gradients of reactive helicity.

Beyond their fundamental significance, these two types of torques may hold the potential to create ultrafast mechanical rotors with a high number of degrees of freedom, of application in torque sensing, gyroscope technologies and the test of fundamental theories of physics, such as Casimir effects and quantum friction[3,28–31]. Overall, we believe that these two identified torques will play a central role in the field of optical manipulation, potentially advancing rotational optomechanics to the Mie-tronics and plasmonics regimes[64,65]. By stepping into the high-order multipole scope, this work also suggests research in the realm of acoustic, hydrodynamic and matter wave manipulation. At last, we would remark that the conservative (or restoring) torque also calls for great attention, as it may prevail for highly anisotropic particles (Fig. 3d). This torque was shown to be a simple harmonic function of $\alpha$ for dipolar particles in usual Gaussian beams[28,52], but it is obvious that an anharmonicity is available for generic higher multipoles and structured light, as shown in Fig. 3e and Eq. (14). Although this restoring torque is beyond the scope of this paper, we anticipate that its nonharmonic property could be exploited in torsional optomechanics[28,52] at the nonlinear regime[66,67].

## Methods

### Nonconservative and conservative optical torques

Torques are essentially different from forces in doing work. The latter does work on a particle by its translational motion (or position change), so nonconservative and conservative forces can be classified according to their divergenceless and irrotational properties[68,69]. However, such a classification method makes no sense to the torques, which do work on the particle by the change in its orientation. For simplicity we shall restrict our analysis to $T_z(\alpha)$, i.e., the $z$-axis rotation; the rotation problems in other directions are similar.

Note that $T_z(\alpha)$ on an arbitrary particle must be a periodic function of $\alpha$: $T_z(\alpha) = T_z(\alpha + 2\pi)$, because $\alpha$ and $\alpha + 2\pi$ describe the same physical configuration; this does not depend on the symmetry of particle property. Therefore, in the spirit of Fourier expansion, the torque can be expressed in a Fourier series:

$$T_z(\alpha) = A_0 + \sum_{n=1}^{\infty} [A_n \sin(n\alpha) + B_n \cos(n\alpha)], \qquad (11)$$

where

$$A_0 = \frac{1}{2\pi} \int_0^{2\pi} T_z(\alpha) d\alpha, \qquad (12)$$

and

$$A_n = \frac{1}{\pi} \int_0^{2\pi} T_z(\alpha) \sin(n\alpha) d\alpha,$$
$$B_n = \frac{1}{\pi} \int_0^{2\pi} T_z(\alpha) \cos(n\alpha) d\alpha. \qquad (13)$$

Eq. (11) is applicable to arbitrary particles. For the dimer with long axis aligning in the $x - y$ plane, we have $T_z(\alpha) = T_z(\alpha + \pi)$ for symmetry, then the odd-$n$ terms in Eq. (11) can be dropped. We argue that the $\alpha$-dependent terms in Eq. (11) make the conservative contribution to the torque:

$$T_z^{\mathrm{cons}}(\alpha) = \sum_{n=1}^{\infty} [A_n \sin(n\alpha) + B_n \cos(n\alpha)], \qquad (14)$$

as it does no work on a particle rotating a full cycle: $\int_0^{2\pi} T_z^{\mathrm{cons}}(\alpha) d\alpha = 0$. For this reason, the particle cannot be rotated continuously under the

action of only $T_z^{\mathrm{cons}}$, and thus this torque is responsible for the orientational restoring (or the locked state). Quite naturally, the $\alpha$-independent leading term (direct-current signal) in Eq. (11) should be nonconservative:

$$T_z^{\mathrm{ncons}} = A_0, \qquad (15)$$

which tends to cause continuous rotation (or the running state), tilting the angular washboard potential[52].

To avoid confusion, we would mention that we have used several classification approaches for the optical torques, which are from different perspectives. Particularly, Eq. (3) classifies the torques by their origins traceable to the properties of illumination. The extinction-scattering decomposition (see Supplementary Eq. (2)) emphasizes their differences associated with the light-matter interaction process. The LOT and NOT feature the unique directions of the torques, while the conservative and nonconservative torques, Eqs. (14) and (15), highlight their different (torsional and rotational) functions in the dynamic control.

## Data availability

The authors declare that all relevant data are available in the paper and Supplementary Information, or from the corresponding author on request.

## Code availability

The code used in this work is available from the corresponding authors upon request.

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

## Acknowledgements

We acknowledge support from the National Natural Science Foundation of China (Nos. 12274181, 12127805, 62135005, 11974417, and 12074169), National Key R&D Program of China (2023YFF0613700), Guangdong Basic and Applied Basic Research Foundation (Nos. 2023A1515030143), and Ministerio de Ciencia, Innovación y Universidades of Spain (Grant PID2022-137569NB-C41).

## Author contributions

X.X., S.Y., and B.Y. designed the research. X.X. and S.Y. developed the theory with input from M.N.V. and F.J.R.F.; Y.Zhou, Y.Zhang, and M.L. were involved in the discussion and analysis. X.X. conceptualized this work. All authors contributed to the manuscript.

## Competing interests

The authors declare no competing interests.
