## [Peer Review File · Nature Communications]

Gradient and curl optical torquesREVIEWER COMMENTS

Reviewer #1 (Remarks to the Author):

Report on the paper 481537_0 “Gradient and curl optical torques” by Xiaohao Xu et al.

The reviewed paper presents a theoretical investigation of the mechanical action performed by structured light fields on small spherical particles. In contrast to the most of other works, it is concentrated on the optical torque which induces the particle spinning near its own axis. So far, only one factor was considered which induces the particle spinning: absorption of light carrying the optical angular momentum (AM); even the method for measuring small light absorption was proposed, based on the observation of light-induced particle spinning, <https://doi.org/10.1364/OE.23.007152>; <https://doi.org/10.1364/AO.54.00F174>. The authors discover additional factors able to induce the particle spinning motion in highly inhomogeneous optical fields: the reactive helicity gradient and the curl of the Poynting momentum. The actions of these factors are manifested through higher multipoles of the Lorentz – Mie expansion but, according to the presented estimations, can be observed and used in the optical-manipulation purposes. It is especially important, that these new factors are able to excite the particle spinning non-collinear with the incident-light AM (lateral optical torque) and even to act oppositely (negative optical torque).

The new data obtained in the paper disclose new features of structured light fields and new possibilities in the optical manipulations.

The paper is written clearly and in the rigorous scientifically sound style (some remarks see below). I did not check the complex calculations but the results look reasonable. I think, it will be interesting and useful for a wide audience, provided that the proper corrections are made before publication. Generally, this work makes an essential new contribution in the fields of opto-mechanics and structured-light properties.

1. The authors warn that they consider “interaction of monochromatic waves with an isotropic and neutral sphere”, after which the symbols “ ω ” and “ k ” are used without explanation. Of course, these are standard notations of frequency and wavenumber but, nevertheless, certain variations in their definitions are possible; these variations (or their absence) should be clearly exposed. The necessity in formal definition is especially evident when the standard symbols are used in non-traditional way: for example, is “ k ” in Eq. (9) the same wavenumber $k = \omega/c$ as in Eq. (4), etc.?

2. Eq. (9) still invokes additional questions. If \mathbf{u} denotes “the state of polarization” and is a transverse vector (a few lines below the case “ $\mathbf{u} = \mathbf{e}_x$ ” is discussed), why does this expression omit the multiplier $\exp(ik_z z)$, which is present, e.g., in (5)? Besides, the meaning of “ ρ ” should be explained immediately, not after two paragraphs.

3. A few lines below Eq. (9) one reads: “one may write the field in the focal plane, by neglecting the other two orthogonal components”  what are the “two orthogonal components” mentioned in this fragment?

4. P. 3, col. 1: “retain the symmetry descending from the Montonen–Olive duality”  It seems that the Montonen–Olive duality is not so well known for the community that needs no explanation; at least, a proper reference should be provided.

Also, I have noticed some small grammar and typesetting issues:

P. 4, col. 1: “which exhibit the same x - and y -dependent properties”  “which exhibits the same x - and y -dependent properties”?

P. 5, col. 1: “The lowest-order multipoles required for the LOF are identified”  “The lowest-order multipoles required for the LOT are identified”?

P. 5, col. 2: “the sign of the LOF is switchable”  “the sign of the LOT is switchable”?

P. 6, col. 1: “It follows direct from the curl component”  “It follows directly from the curl component”?

P. 6, col. 2: “Poynting momentum-curl”  “Poynting-momentum curl”?

Section “Discussion and conclusion” starts with the statement “We have built a multipole theory for the optical torque, with a classifying framework featuring two fundamental field properties” but a few lines below one reads: “First, these three aforementioned quantities...”. Do the authors mean that the “aforementioned” spin is also included to their “classifying framework”? Anyway, this fragment should be presented more clearly.

Conclusion: the manuscript can be accepted after minor corrections.

Reviewer #2 (Remarks to the Author):

The manuscript by Xu and coworkers presents a general framework for calculating and understanding the torque exerted on a Mie particle by an arbitrary electromagnetic field. The framework allows to calculate torques in general case, where electromagnetic field carries both SAM and OAM, and the particles are described in terms of multipolar expansion. The manuscript shows that for multipoles higher than a dipole, torques from helicity gradient and momentum-curl of light arise and discuss the scenarios in which negative optical torque (NOT) and lateral optical torque (LOT) are created (using structured light beams).

The authors claim to “theoretically uncover and predict the existence of two fundamental torque components”, which I believe is exaggerated. The torque terms proportional to helicity gradient and curl of momentum are already present in e.g. Phys. Rev. A 92, 043843. The novelty lies in the realization that these terms can (under idealized conditions) produce LOT.

The manuscript is clear and well written, and the presented theoretical torque framework is mathematically very elegant. However, I believe its relevance for experimentalists is limited, since only ideal spheres are considered. Since the LOT and NOT are only present for larger Mie particles (outside dipolar approximation), considering anisotropic particles is very important. Some anisotropy is necessary so that the rotation due to torque is detectable, and the anisotropy-related effects such as restoring torques and scattering torques will likely prevail in experiments. Therefore, including the particles anisotropy is necessary to provide a truly realistic pathway towards observing LOT in experiment.

Finally, the discussion of the torque in the optical spanner appears disjointed from the remainder of the manuscript. I would suggest to omit the “optical spanner” section, so that the manuscript is more focused.

Overall, I miss the broader relevance and the groundbreaking character that would make the manuscript also interesting to experimentalists. Such broader relevance could for instance be expanding the framework to include non-spherical particles. In addition, I am

not convinced LOT is interesting/relevant. I therefore cannot recommend publication of the manuscript. It may be better suited to a more specialized journal.

More detailed remarks:

1. When describing the Eq. (1) the authors only speak about the incident field. I find the discussion of the interaction of the particle with its own scattered field missing. In realistic settings (e.g. Phys. Rev. Lett. 129, 023602) interaction with scattered field leads to torques larger than absorption.
2. It would be interesting to include the discussion of the angular momentum conservation in the context of LOT.

Response to Reviewer #1

The reviewed paper presents a theoretical investigation of the mechanical action performed by structured light fields on small spherical particles. In contrast to the most of other works, it is concentrated on the optical torque which induces the particle spinning near its own axis. So far, only one factor was considered which induces the particle spinning: absorption of light carrying the optical angular momentum (AM); even the method for measuring small light absorption was proposed, based on the observation of light-induced particle spinning, <https://doi.org/10.1364/OE.23.007152>; <https://doi.org/10.1364/AO.54.00F174>. The authors discover additional factors able to induce the particle spinning motion in highly inhomogeneous optical fields: the reactive helicity gradient and the curl of the Poynting momentum. The actions of these factors are manifested through higher multipoles of the Lorentz – Mie expansion but, according to the presented estimations, can be observed and used in the optical-manipulation purposes. It is especially important, that these new factors are able to excite the particle spinning non-collinear with the incident-light AM (lateral optical torque) and even to act oppositely (negative optical torque).

The new data obtained in the paper disclose new features of structured light fields and new possibilities in the optical manipulations. The paper is written clearly and in the rigorous scientifically sound style (some remarks see below). I did not check the complex calculations but the results look reasonable. I think, it will be interesting and useful for a wide audience, provided that the proper corrections are made before publication. Generally, this work makes an essential new contribution in the fields of optomechanics and structured-light properties.

Response (R1.1): We are grateful to the referee’s high judgment of our manuscript. The corrections are addressed as follows.

1. The authors warn that they consider “interaction of monochromatic waves with an isotropic and neutral sphere”, after which the symbols “ ω ” and “ k ” are used without explanation. Of course, these are standard notations of frequency and wavenumber but, nevertheless, certain variations in their definitions are possible; these variations (or their absence) should be clearly exposed. The necessity in formal definition is especially evident when the standard symbols are used in non-traditional way: for example, is “ k ” in Eq. (9) the same wavenumber $k = \omega/c$ as in Eq. (4), etc.?

R1.2: Yes, the wave numbers in Eqs. (9) and (4) are the same. We explain k and ω in the revision (please see **P. 2, col. 1**).

2. Eq. (9) still invokes additional questions. If \mathbf{u} denotes “the state of polarization” and is a transverse vector (a few lines below the case “ $\mathbf{u} = \mathbf{e}_x$ ” is discussed), why does this expression omit the multiplier $\exp(ik_z z)$, which is present, e.g., in (5)? Besides, the meaning of “ ρ_0 ” should be explained immediately, not after two paragraphs.

R1.3: Eq. (9) is the expression for the field at the pupil plane $z = z_0$, so the propagating phase term $\exp(ik_z z_0)$, which is a constant, can be omitted. Eq. (5) formulates the field

in the whole 3D space, so the variable phase term $\exp(ik_z z)$ cannot be dropped. We explain the meaning of “ ρ_0 ” immediately following Eq. (9) in the revision (P. 6, col. 2).

3. A few lines below Eq. (9) one reads: “one may write the field in the focal plane, by neglecting the other two orthogonal components”  what are the “two orthogonal components” mentioned in this fragment?

R1.4: We refer to the two components orthogonal to the polarization \mathbf{u} . For example, if \mathbf{u} represents the left-handed circular polarization $|e_L\rangle$, then the two orthogonal components are the longitudinal polarization and right-handed circular polarization $|e_R\rangle$. In the revision, we rephrase the related statement for clarity (P. 6, col. 2).

4. P. 3, col. 1: “retain the symmetry descending from the Montonen–Olive duality”  It seems that the Montonen–Olive duality is not so well known for the community that needs no explanation; at least, a proper reference should be provided.

R1.5: We thank the referee for this important comment. What the “Montonen–Olive duality” actually refers to is the dual symmetry. However, we now realize the spin torque in its form does not follow this symmetry, as it is asymmetric with respect to the exchange of the electric and magnetic vectors: $\mathbf{E} \rightarrow \mathbf{B}c$, $\mathbf{B}c \rightarrow -\mathbf{E}$. We ignored the difference in the prefactors of the electric and magnetic parts of the torque. For these reasons, this statement is removed in the revision.

Also, I have noticed some small grammar and typesetting issues:

P. 4, col. 1: “which exhibit the same x - and y -dependent properties”  “which exhibits the same x - and y -dependent properties”?

R1.6: Thanks for your correction. The issue is fixed.

P. 5, col. 1: “The lowest-order multipoles required for the LOF are identified”  “The lowest order multipoles required for the LOT are identified”?

R1.7: Corrected.

P. 5, col. 2: “the sign of the LOF is switchable”  “the sign of the LOT is switchable”?

R1.8: Corrected.

P. 6, col. 1: “It follows direct from the curl component”  “It follows directly from the curl component”?

R1.9: Corrected.

P. 6, col. 2: “Poynting momentum-curl”  “Poynting-momentum curl”?

R1.10: Corrected.

Section “Discussion and conclusion” starts with the statement “We have built a multipole theory for the optical torque, with a classifying framework featuring two

fundamental field properties” but a few lines below one reads: “First, these three aforementioned quantities...”. Do the authors mean that the “aforementioned” spin is also included to their “classifying framework”? Anyway, this fragment should be presented more clearly.

R1.11: We thank the referee for raising this point. Yes, the spin is also included in our classifying framework. In the revised manuscript, we rephrase the related sentence as “*We have built a multipole theory for the optical torque, with a classifying framework featuring three fundamental field properties: the optical spin...*”.

Conclusion: the manuscript can be accepted after minor corrections.

Response to Reviewer #2

The manuscript by Xu and coworkers presents a general framework for calculating and understanding the torque exerted on a Mie particle by an arbitrary electromagnetic field. The framework allows to calculate torques in general case, where electromagnetic field carries both SAM and OAM, and the particles are described in terms of multipolar expansion. The manuscript shows that for multipoles higher than a dipole, torques from helicity gradient and momentum-curl of light arise and discuss the scenarios in which negative optical torque (NOT) and lateral optical torque (LOT) are created (using structured light beams).

Response (R2.1): We thank the referee for his/her time and effort invested in reading and evaluating our manuscript. We also thank the referee for the constructive comments, which helped a lot to improve our manuscript.

The authors claim to “theoretically uncover and predict the existence of two fundamental torque components”, which I believe is exaggerated. The torque terms proportional to helicity gradient and curl of momentum are already present in e.g. Phys. Rev. A 92, 043843. The novelty lies in the realization that these terms can (under idealized conditions) produce LOT.

R2.2: We apologize for any confusion caused. Concerning the paper by one of us, Nieto-Vesperinas, PRA 92, 043843 (2015), quoted as Ref. [41] in the previous manuscript (now Ref. [40]), there are fundamental differences between this manuscript and that PRA paper.

The work of that PRA paper was restricted to dipolar particles. It is true that Nieto-Vesperinas resorted to the gradient and curl terms, in deriving the torque contributions from the conservation of spin (SAM) and of orbital angular momenta (OAM), separately. However, when one adds these SAM and OAM, thus taking fully into account the conservation of total AM, (which represents the physical observable), no such gradient and curl components appear, because they cancel out in that addition that yields the net torque on the dipolar particle, which in this dipole approximation only depends on the optical spin and the absorption cross-section of the dipolar particle, as shown in Eq. (1), as well as Refs. [17,18,41] quoted in the revision.

In our manuscript, multipoles of arbitrary order are addressed, and we show that the gradient and curl components are present in the net torque on multipoles higher than the dipole order, so they can be said to exist. In the revision, we remark on the results in Ref. [40] to further clarify the novelty of our paper (please see P. 3, col. 2).

The manuscript is clear and well written, and the presented theoretical torque framework is mathematically very elegant. However, I believe its relevance for experimentalists is limited, since only ideal spheres are considered. Since the LOT and NOT are only present for larger Mie particles (outside dipolar approximation), considering anisotropic particles is very important. Some anisotropy is necessary so that the rotation due to torque is detectable, and the anisotropy-related effects such as restoring torques and scattering torques will likely prevail in experiments. Therefore, including the particles anisotropy is necessary to provide a truly realistic pathway towards observing LOT in experiment.

R2.3: Thank you for your recognition of our theoretical framework. We also agree with you that particles in reality cannot be perfect spheres and the restoring torque could be significant for some highly anisotropic structures. In response to this referee's concern, we have added completely new section, where we perform numerical simulation on anisotropic spheres, including roughness, as well as anisotropic dimers of two spheres. As regards the scattering torque mentioned by the reviewer, we have in fact considered this contribution in our theory. In particular, our framework is derived from Supplementary Eq. (1), in which $\mathbf{T}_{\text{sca}}^{(l)}$ accounts for the scattering (or recoiling) torque due to the l -order multipoles. In the following reply, we shall focus on the relevance of our theory to realistic spherical particles and nonspherical structures, by comparing it with numerical results and by the method of T matrix.

To simulate the realistic sphere-like particles, we consider spheres of varying roughness (please see particles i-iv in Fig. R1a), with the illumination of the traveling wave as an example. The optical torque is computed based on Maxwell stress tensor method, in which the electromagnetic field is obtained by finite-difference time-domain (FDTD) simulations. Figure R1b shows the calculated LOT (i.e., T_z) on the particles with varying orientation angle α at $y = 0$ and $x = 0.35\lambda$, where the momentum curl of illumination is maximized. The α -independent torque on the isotropic particle i is approximately $-0.1 \text{ pN}\cdot\mu\text{m}$, in good agreement with our analytical theory (please see red line in Fig. 2d in the manuscript). Due to the anisotropy, the torque magnitude varies with α for the rough spheres ii-iv, and larger roughness yields more significant variation. However, the torque maintains its negative sign, irrespective to the orientation. This indicates that the rough spheres will exhibit the running state to rotate clockwise and continuously, consistent with the perfect sphere, as shown by the angular potential (Fig. R1c), which is calculated by the opposite angular integral of the torque: $-\int_0^\alpha T_z d\alpha$. It also suggests that the restoring contribution of the torque is overwhelmed by that responsible for continuous rotation. Therefore, our theory can well predict and explain the dynamic behaviors of these sphere-like structures.

Fig. R1. FDTD simulation results for different shaped particles. **a** Sectional view of particles used in the simulations. Particle i represents an ideal sphere with radius of $0.1 \mu\text{m}$; particles ii-iv are rough spheres modeled by varying radius values, which converge in distribution to a Gaussian variable with mean $0.1 \mu\text{m}$ and standard deviation $\sigma = 1, 2$ and 3 nm , respectively; particles v and vi are dimers composed of dual spheres with radius of 0.1 and $0.05 \mu\text{m}$, respectively. **b** LOT as a function of the orientation angle α of the spherical particles i-iv. The angle is defined with respect to x -axis, as illustrated by the inset. **c** Calculated angular potential for the particles i-iv. **d** LOT and its conservative and nonconservative components versus the orientation angle of the dimer v. **e** A similar plot for the dimer vi. **f** Angular potential for v and vi. In all simulations, the particles are illuminated by the travelling wave (case I) in Fig. 2a, and are placed at a fixed position $x = 0.35\lambda$, where the momentum curl of illumination reaches the maximum.

We then evaluate the LOT and angular potential for the dimers (i.e., particles v and vi in Fig. R1a) made of two identical Si spheres of radius r . For this highly anisotropic geometry, it is instructive to extract the nonconservative and conservative (or restoring) parts from the torque:

$$T_z(\alpha) = T_z^{\text{ncons}} + T_z^{\text{cons}}(\alpha), \quad (\text{R1})$$

where

$$T_z^{\text{ncons}} = \frac{1}{2\pi} \int_0^{2\pi} T_z(\alpha) d\alpha, \quad T_z^{\text{cons}}(\alpha) = T_z(\alpha) - T_z^{\text{ncons}}. \quad (\text{R2})$$

The nonconservative α -independent part T_z^{ncons} tends to cause continuous rotation (or the running state), tilting the angular washboard potential. The conservative part $T_z^{\text{cons}}(\alpha)$, by definition, does no work on the particle rotating a full cycle:

$$\int_0^{2\pi} T_z^{\text{cons}}(\alpha) d\alpha = 0, \text{ so the particle cannot be rotated continuously under the action of}$$

only this torque. In this sense, $T_z^{\text{cons}}(\alpha)$ is responsible for the orientational restoring (or the locked state).

The results for the smaller dimer (particle v) are shown in Fig. R1d. The total LOT, T_z , varies with the orientation angle in a sinusoidal-like way, and it changes sign with a negative derivative around $\alpha = 90^\circ$ or 270° . Accordingly, an angular potential well is formed, by which the particle exhibits the locked (torsional) state (Fig. R1f). It is also noted that the washboard potential is slightly tilted due to the presence of a small nonconservative component T_z^{nccons} in Fig. R1d. However, the dynamic behavior of the larger dimer (particle vi) is quite different. As shown in Fig. R1e, T_z is always negative, because the magnitude of T_z^{nccons} dominates over that of T_z^{cons} . As a result, the potential well vanishes, switching the dimer to the running state (Fig. R1f).

Conventionally, the running state and washboard potential tilt are attributed to the circular polarization (or optical spin) of light [Phys. Rev. Lett. 2022, 129, 023602 (quoted as Ref. [51] in revised paper)]. They are, however, explained by the momentum curl in our case, because only this quantity of illumination has the lateral component (see the subsequent paragraph for detailed proof). It also interprets the dominant restoring effect on the small dimer, because the momentum curl produces torque through higher multipole responses, which are weak for reduced particle size.

We now present a rigorous argument proving that the nonconservative component T_z^{nccons} in Fig. R1, which causes the running state and washboard potential tilt, is traced to the momentum curl. For a dipolar dimer or other rotationally symmetrical structures (e.g., cylinders and spheroids), the torque can be derived by replacing the scalar polarizability, i.e., $\gamma_{e(m)}^{(l=1)}$ in Supplementary Eq. (3), by a tensor, as previous work demonstrated [Phys. Rev. Lett. 2022, 129, 023602]. However, such a treatment will be invalid for anisotropic higher multipoles, because it will break the symmetric and traceless properties of multipole moments, which are required by definition [J. D. Jackson, Classical Electrodynamics (Wiley, New York, 1962)]. To address the anisotropy, we employ a generic expression of the torque [JOSA A, 2007, 24(2): 430-443]:

$$T_z = -\frac{\mathcal{E}}{2k^3} \sum_{l=1}^N \sum_{m=-l}^l m \left[\text{Re}(q_{ml} b_{ml}^* + p_{ml} a_{ml}^*) + |q_{ml}|^2 + |p_{ml}|^2 \right], \quad (\text{R3})$$

where p_{ml} and q_{ml} are the expansion coefficients of the scattered field on the basis of vector spherical wave functions (VSWFs). They are linked to the expansion coefficients of the incident field, a_{ml} and b_{ml} , via the T matrix:

$$\begin{aligned} p_{ml} &= \sum_{l'=1}^N \sum_{m'=-l'}^{l'} (T_{mlm'l'}^{11} a_{m'l'} + T_{mlm'l'}^{12} b_{m'l'}), \\ q_{ml} &= \sum_{l'=1}^N \sum_{m'=-l'}^{l'} (T_{mlm'l'}^{21} a_{m'l'} + T_{mlm'l'}^{22} b_{m'l'}). \end{aligned} \quad (\text{R4})$$

Note that in Eq. (R3) the terms related to $\text{Re}(q_{ml} b_{ml}^* + p_{ml} a_{ml}^*)$ and $|q_{ml}|^2 + |p_{ml}|^2$

correspond to the extinction and scattering torques in Supplementary Eq. (2), respectively. Each element of the T matrix $\mathbf{T} = \begin{bmatrix} \mathbf{T}^{11} & \mathbf{T}^{12} \\ \mathbf{T}^{21} & \mathbf{T}^{22} \end{bmatrix}$ has the following general symmetric property,

$$T_{-ml, -m'l'}^{ij} = (-1)^{m+m'} T_{m'l, ml}^{ji}. \quad (\text{R5})$$

Substituting Eq. (R4) into (R3) yields

$$\begin{aligned} T_z = & -\frac{\varepsilon}{2k^3} \sum_{l=1}^N \sum_{m=-l}^l \sum_{l'=1}^N \sum_{m'=-l'}^{l'} m \operatorname{Re} \left(T_{mlm'l'}^{21} a_{m'l} b_{ml}^* + T_{mlm'l'}^{22} b_{m'l} b_{ml}^* + T_{mlm'l'}^{11} a_{m'l} a_{ml}^* + T_{mlm'l'}^{12} b_{m'l} a_{ml}^* \right) \\ & - \frac{\varepsilon}{2k^3} \sum_{l=1}^N \sum_{m=-l}^l \sum_{l'=1}^N \sum_{m'=-l'}^{l'} \left(S_{mlm'l'}^{21} a_{m'l} b_{ml}^* + S_{mlm'l'}^{22} b_{m'l} b_{ml}^* + S_{mlm'l'}^{11} a_{m'l} a_{ml}^* + S_{mlm'l'}^{12} b_{m'l} a_{ml}^* \right), \end{aligned} \quad (\text{R6})$$

where $S_{mlm'l'}^{ij}$ represents the element of the matrix constructed by

$$\mathbf{S} = \begin{bmatrix} \mathbf{S}^{11} & \mathbf{S}^{12} \\ \mathbf{S}^{21} & \mathbf{S}^{22} \end{bmatrix} = \mathbf{T}^\dagger \mathbf{M} \mathbf{T}, \quad M_{mlm'l'} = m \delta_{mm'} \delta_{ll'}. \quad (\text{R7})$$

We notice that in Eq. (R6) only $T_{mlm'l'}^{ij}$ and $S_{mlm'l'}^{ij}$ depend on the particle's orientation α .

Then performing the angular integral or average over α for this equation, we have the nonconservative torque given by:

$$\begin{aligned} T_z^{\text{ncons}} = \langle T_z \rangle = & -\frac{\varepsilon}{2k^3} \sum_{l=1}^N \sum_{m=-l}^l \sum_{l'=1}^N \sum_{m'=-l'}^{l'} m \operatorname{Re} \left(\langle T_{mlm'l'}^{21} \rangle a_{m'l} b_{ml}^* + \langle T_{mlm'l'}^{22} \rangle b_{m'l} b_{ml}^* \right. \\ & \left. + \langle T_{mlm'l'}^{11} \rangle a_{m'l} a_{ml}^* + \langle T_{mlm'l'}^{12} \rangle b_{m'l} a_{ml}^* \right) \\ & - \frac{\varepsilon}{2k^3} \sum_{l=1}^N \sum_{m=-l}^l \sum_{l'=1}^N \sum_{m'=-l'}^{l'} \left(\langle S_{mlm'l'}^{21} \rangle a_{m'l} b_{ml}^* + \langle S_{mlm'l'}^{22} \rangle b_{m'l} b_{ml}^* \right. \\ & \left. + \langle S_{mlm'l'}^{11} \rangle a_{m'l} a_{ml}^* + \langle S_{mlm'l'}^{12} \rangle b_{m'l} a_{ml}^* \right) \end{aligned} \quad (\text{R8})$$

where $\langle \cdot \rangle$ denotes the angular average operator. Utilizing Eq. (R5), the Wigner D -functions (see Ref. [53]) and the identity, $\int_0^{2\pi} e^{-i(m-m')\alpha} d\alpha = \delta_{mm'}$, one may obtain the following characteristics for the averaged matrix elements:

$$\begin{aligned} \langle T_{mlm'l'}^{ij} \rangle & = \delta_{mm'} \langle T_{mlm'l'}^{ij} \rangle, \quad \langle T_{-ml, -m'l'}^{ij} \rangle = (-1)^{i+j} \langle T_{mlm'l'}^{ij} \rangle, \\ \langle S_{mlm'l'}^{ij} \rangle & = \delta_{mm'} \langle S_{mlm'l'}^{ij} \rangle, \quad \langle S_{-ml, -m'l'}^{ij} \rangle = (-1)^{i+j+1} \langle S_{mlm'l'}^{ij} \rangle. \end{aligned} \quad (\text{R9})$$

With Eqs. (R9), we can reduce the summations in (R8) and recast it into a more symmetric form:

$$T_z^{\text{ncons}} = -\frac{\varepsilon}{2k^3} \sum_{l=1}^N \sum_{m=0}^l \sum_{l'=1}^N \left(1 - \frac{\delta_{m0}}{2} \right) \operatorname{Re} \left[\begin{aligned} & \left(m \langle T_{mlm'l'}^{21} \rangle + \langle S_{mlm'l'}^{21} \rangle \right) (a_{m'l} b_{ml}^* + a_{-ml} b_{-ml}^*) \\ & + \left(m \langle T_{mlm'l'}^{12} \rangle + \langle S_{mlm'l'}^{12} \rangle \right) (b_{m'l} a_{ml}^* + b_{-ml} a_{-ml}^*) \\ & + \left(m \langle T_{mlm'l'}^{22} \rangle + \langle S_{mlm'l'}^{22} \rangle \right) (b_{m'l} b_{ml}^* - b_{-ml} b_{-ml}^*) \\ & + \left(m \langle T_{mlm'l'}^{11} \rangle + \langle S_{mlm'l'}^{11} \rangle \right) (a_{m'l} a_{ml}^* - a_{-ml} a_{-ml}^*) \end{aligned} \right]. \quad (\text{R10})$$

Then the key step is to express a_{ml} and b_{ml} in terms of the incident fields. This can

be done by taking advantage of the orthogonality among vector spherical harmonic functions (VSHFs) (page 377, Ref.[53]):

$$a_{ml} = 4\pi(-1)^m i^l G_{ml} \mathbf{E} \cdot \mathbf{C}_{-ml}(\theta, \phi), \quad b_{ml} = 4\pi(-1)^m i^{l-1} G_{ml} \mathbf{E} \cdot \mathbf{B}_{-ml}(\theta, \phi), \quad (\text{R11})$$

where

$$G_{ml} = \sqrt{\frac{(2l+1)(l+m)!}{4\pi(l+1)(l-m)!}},$$

and the VSHFs are given by

$$\begin{aligned} \mathbf{B}_{lm}(\theta, \phi) &= \left[\mathbf{e}_\theta \frac{d}{d\theta} P_l^m(\cos \theta) + \mathbf{e}_\phi \frac{im}{\sin \theta} P_l^m(\cos \theta) \right] e^{im\phi}, \\ \mathbf{C}_{lm}(\theta, \phi) &= \left[\mathbf{e}_\theta \frac{im}{\sin \theta} P_l^m(\cos \theta) - \mathbf{e}_\phi \frac{d}{d\theta} P_l^m(\cos \theta) \right] e^{im\phi}. \end{aligned}$$

Plugging Eq. (5) in the manuscript into (R11) yields the formulas:

$$\begin{aligned} a_{ml} &= 2\pi E_0 (-1)^m i^l G_{ml} \left[i(e^{iKx} - e^{-im\pi} e^{-iKx}) \pi_{-m,l} + \cos \theta (e^{-im\pi/2} e^{iKy} + e^{im\pi/2} e^{-iKy}) \tau_{-m,l} \right] e^{ik_z z}, \\ b_{ml} &= 2\pi E_0 (-1)^m i^{l-1} G_{ml} \left[(e^{iKx} - e^{-im\pi} e^{-iKx}) \tau_{-m,l} - i \cos \theta (e^{-im\pi/2} e^{iKy} + e^{im\pi/2} e^{-iKy}) \pi_{-m,l} \right] e^{ik_z z}, \end{aligned} \quad (\text{R12})$$

with

$$\tau_{ml} = \frac{d}{d\theta} P_l^m(\cos \theta), \quad \pi_{ml} = \frac{m}{\sin \theta} P_l^m(\cos \theta).$$

Ultimately, with Eqs. (R12), the identity for the associated Legendre polynomials $P_l^{-m}(\cos \theta) = (-1)^m \frac{(l-m)!}{(l+m)!} P_l^m(\cos \theta)$, and Eq. (7) in the manuscript, we are led to the following results for the traveling wave (i.e., real k_z):

$$\begin{cases} \begin{bmatrix} a_{ml} b_{ml}^* + a_{-ml} b_{-ml}^* \\ b_{ml} a_{ml}^* + b_{-ml} a_{-ml}^* \\ b_{ml} b_{ml}^* - b_{-ml} b_{-ml}^* \\ a_{ml} a_{ml}^* - a_{-ml} a_{-ml}^* \end{bmatrix} \\ \left\{ \begin{array}{l} \propto \sin(Kx) \cos(Ky) \propto (\nabla \times \mathbf{p})_z \quad \text{for even } m \\ 0 \quad \text{for odd } m \end{array} \right. \end{cases} \quad (\text{R13})$$

It follows from Eqs. (R10) and (R13) that T_z^{ncons} is proportional to the momentum curl $(\nabla \times \mathbf{p})_z$ for higher multipoles ($N > 1$). However, T_z^{ncons} should be zero on the dipoles ($N = 1$), for which $l = l' = 1$ and $m = 0, 1$ in Eq. (R10). This is because the matrix elements-related terms in Eq. (R10) are zero at $l = l' = 1$ and $m = 0$; and for $m = 1$ (odd) the expansion coefficients-related terms will vanish according to Eq. (R13).

On the other hand, one may also work out the expansion coefficients-related terms for the evanescent wave (i.e., imaginary $k_z = iq$):

$$\begin{cases} \begin{bmatrix} a_{ml} b_{ml}^* + a_{-ml} b_{-ml}^* \\ b_{ml} a_{ml}^* + b_{-ml} a_{-ml}^* \\ b_{ml} b_{ml}^* - b_{-ml} b_{-ml}^* \\ a_{ml} a_{ml}^* - a_{-ml} a_{-ml}^* \end{bmatrix} \\ \left\{ \begin{array}{l} \propto \sin(Kx) \cos(Ky) e^{-2qz} \propto (\nabla \mathcal{H})_z \quad \text{for even } m \\ 0 \quad \text{for odd } m \end{array} \right. \end{cases} \quad (\text{R14})$$

Therefore, the nonconservative torque T_z^{ncons} on the higher multipoles is induced by the

reactive helicity gradient $(\nabla\mathcal{H})_z$ for the evanescent wave. Likewise, this gradient torque should also be zero for the dipoles.

The above results regarding the anisotropic particles are incorporated into the manuscript (please see new section “LOT on anisotropic particles” and Fig. 3) and Supplementary Information (Section B). We also add a Method section to elaborate the physics of the conservative and nonconservative torques, from the viewpoint of Fourier series expansion.

Finally, the discussion of the torque in the optical spanner appears disjointed from the remainder of the manuscript. I would suggest to omit the “optical spanner” section, so that the manuscript is more focused.

R2.4: We thank the referee for his/her suggestion, but the “optical spanner” section indeed fits well with the principal line of the manuscript. Our work focuses on both the gradient and curl components of the optical torque, and we demonstrate in this section that the curl component is of relevance in the optical spanner configuration. Particularly, this curl component is utilized to realize the negative optical torque (NOT), which is considered “*especially important*” by the other reviewer. Although the NOT was realized by taking advantage of anisotropic or inhomogeneous materials (cf. Ref. [32-39]), we demonstrate here for the first time that this phenomenon is also possible for the homogeneous isotropic sphere. For these reasons, we would reserve this section in the revised manuscript.

Overall, I miss the broader relevance and the groundbreaking character that would make the manuscript also interesting to experimentalists. Such broader relevance could for instance be expanding the framework to include non-spherical particles. In addition, I am not convinced LOT is interesting/relevant. I therefore cannot recommend publication of the manuscript. It may be better suited to a more specialized journal.

R2.3: We would mention that spherical particles are of interest to experimentalists. For example, in a recent paper [*Nature Nanotechnology* 2020, 15, 89–93], the authors employed both dimers and nanospheres for torque sensing, and they chose the nanospheres to explore the possibility of observing the long-sought-after vacuum friction. The spinning microspheres were also used for observing Magnus effects at small scales [*Nature Physics*, 2023, 19(12): 1904-1909]. It is true that experimentally available particles cannot be perfect spheres, but we have demonstrated numerically that our theory works well for rough spheres.

We have also considered a highly anisotropic geometry (the dimer) and demonstrated that the nonconservative component of torque may overcome the restoring one, spinning the dimer continuously (Figs. R1e and f). We develop a framework (i.e., Eq. (R10)) to evaluate the nonconservative torque, which is shown to be induced by the momentum curl (for the travelling wave, Eq. (R13)) or reactive helicity gradient (for the evanescent wave, Eq. (R14)). Although we do not present in-depth analysis of the restoring or conservative torque, which is beyond the scope of this

paper, we stress in the end of the manuscript the importance of this torque in torsional optomechanics for future outlook (P. 8, col. 2).

As regards the relevance of the LOT, it is embodied in two aspects. For optical-manipulation purposes, the LOT opens the opportunity to drive the particle spinning non-collinear with the incident optical spin. In terms of scientific aesthetics, the LOT sets the rotational analog to the well-known lateral optical force (LOF). Given the surging attention the LOF bagged from the community, as evident from the following publications (to name a few):

- Nature Communications, 2014, 5(1): 3307.
- Nature Communications, 2015, 6(1): 8799.
- PNAS, 2015, 112(43): 13190-13194.
- Nature Photonics, 2018, 12(8): 461-464.
- Physical Review Letters, 2020, 125(7): 073901.
- Science Advances, 2020, 6(45): eabc3726.
- Science Advances, 2022, 8(48): eabn2291.
- Nature Communications, 2023, 14(1): 6361.
- Advances in Optics and Photonics, 2023, 15(3): 835-906.

we believe that the LOT would also draw the interest of a broad readership of Nature Communications.

More detailed remarks:

1. When describing the Eq. (1) the authors only speak about the incident field. I find the discussion of the interaction of the particle with its own scattered field missing. In realistic settings (e.g. Phys. Rev. Lett. 129, 023602) interaction with scattered field leads to torques larger than absorption.

R2.6: We thank the referee for bringing this important reference to our attention, which is quoted as Ref. [51] in the revision. As for the scattering contributions to the torque, we had already considered it in Eq. (1). Please be advised that the electric part of Eq. (1) is equivalent to the optical torque formula in this reference, i.e., Eq. (12) in Supplemental Material of Ref. [51]:

$$\mathbf{M} = \frac{1}{2} \text{Re}(\mathbf{p} \times \mathbf{E}^*) + \frac{k^3}{12\pi\epsilon} \text{Im}(\mathbf{p} \times \mathbf{p}^*). \quad (\text{R15})$$

The first term of Eq. (R15) represents the interaction with the incident field, while the second term depends exclusively on the scattering. In fact, by expressing the electric dipole moment \mathbf{p} in terms of the polarizability $\gamma_e^{(1)}$ and electric vector: $\mathbf{p} = \gamma_e^{(1)}\mathbf{E}$, one may express Eq. (R15) as

$$\begin{aligned}
\mathbf{M} &= \frac{1}{2} \operatorname{Re}(\mathbf{p} \times \mathbf{E}^*) + \frac{k^3}{12\pi\epsilon} \operatorname{Im}(\mathbf{p} \times \mathbf{p}^*) \\
&= \frac{1}{2} \operatorname{Im} \gamma_e^{(1)} \operatorname{Im}(\mathbf{E}^* \times \mathbf{E}) - \frac{k^3}{12\pi\epsilon} |\gamma_e^{(1)}|^2 \operatorname{Im}(\mathbf{E}^* \times \mathbf{E}) \\
&= \frac{c}{2} \left[\frac{k}{\epsilon} \operatorname{Im} \gamma_e^{(1)} - \frac{k^4}{6\pi\epsilon^2} |\gamma_e^{(1)}|^2 \right] \frac{\epsilon}{\omega} \operatorname{Im}(\mathbf{E}^* \times \mathbf{E}) \\
&= \frac{c}{2} [C_{\text{ext-e}}^{(1)} - C_{\text{sca-e}}^{(1)}] \mathbf{s}_e \\
&= \frac{c}{2} C_{\text{abs-e}}^{(1)} \mathbf{s}_e
\end{aligned} \tag{R16}$$

where $\mathbf{s}_e = \frac{\epsilon}{\omega} \operatorname{Im}(\mathbf{E}^* \times \mathbf{E})$ is the electric spin; the difference between the extinction and

scattering cross-sections, $C_{\text{ext-e}}^{(1)} = \frac{k}{\epsilon} \operatorname{Im} \gamma_e^{(1)}$ and $C_{\text{sca-e}}^{(1)} = \frac{k^4}{6\pi\epsilon^2} |\gamma_e^{(1)}|^2$, defines the

absorption cross-section: $C_{\text{abs-e}}^{(1)} = C_{\text{ext-e}}^{(1)} - C_{\text{sca-e}}^{(1)}$. The scattering contribution is reflected

by the term related to the scattering cross-section $C_{\text{sca-e}}^{(1)}$. Consequently, Eq. (R16) is the electric part of our Eq.(1).

2. It would be interesting to include the discussion of the angular momentum conservation in the context of LOT.

R2.7: As per your suggestion, we add the discussion in the manuscript (P. 5, col. 1).

REVIEWERS' COMMENTS

Reviewer #1 (Remarks to the Author):

I support the paper publication. See the attachment please. [attachment below]

Report on the resubmitted paper 481537_1 “Gradient and curl optical torques” by Xiaohao Xu et al.

Upon resubmission, the authors have carefully addressed the previous remarks, and now the paper is essentially improved. It is only one minor point relating the momentum-curl torque (3rd Eq. (3)), that I overlooked in the 1st review. Actually, in some cases the term $\propto \nabla \times \mathbf{p}$, \mathbf{p} being the Poynting vector or a certain its constituent, represents a sort of the “true” electromagnetic spin (see, for example, [47], Transverse spin and the hidden vorticity of propagating light fields <https://doi.org/10.1364/JOSAA.466360>; Transverse spinning of unpolarized light <https://doi.org/10.1038/s41566-020-00733-3>). In such cases, the “momentum-curl” contribution is, at least partially, “captured” by the “pure-spin” torque (1). Accordingly, the “momentum-curl” torque can be “felt” even by dipole particles. Am I right? If so, this point should be briefly commented to avoid confusing ambiguities.

As to the other materials, I can only suggest some small technical corrections:

P. 3, col. 1, line 3 from the bottom: “ $\nabla \times \mathbf{P}$ ”  “ $\nabla \times \mathbf{p}$ ”?

P.3, col. 2, beneath Eq. (6): “four-plane waves”  “four plane waves”?

P. 5, col. 2, line 10: “Lorenz-Mie method-based results”  “results based on the Lorenz-Mie method”?

P. 6, col. 1: “washboard potential is slightly titled”  “washboard potential is slightly tilted”?

P. 6, col. 2: “Richard-Wolf method”  “Richards-Wolf method”?

P. 8, col. 2: “Due to symmetry, $T_z(\alpha)$ on an arbitrary particle must be a periodic function of α ”  $T_z(\alpha)$ must be a periodic function of α because α and $\alpha + 2\pi$ describe the same physical configuration; this does not depend on symmetry.

P. 11: Ref. 62 is incomplete.

I think, the suggested corrections are technical and can be made during the final text polishing. I support the paper publication

Reviewer #2 (Remarks to the Author):

The manuscript by Xu et al. introduces a comprehensive framework for describing the torque exerted on a Mie particle by any electromagnetic field. The study reveals that beyond dipolar approximation, torques originating from helicity gradients and momentum-curl of light emerge. Moreover, it explores situations where structured light beams induce negative optical torque (NOT) and lateral optical torque (LOT).

The authors responded extensively to my queries. My concern regarding the paper's incremental nature compared to prior work by one of the authors (Nieto-Vesperinas, PRA 92, 043843 (2015)) remains valid. However, the manuscript's inclusion of discussions on LOT affecting anisotropic particles significantly enhances its quality and broadens its appeal to the scientific community. Therefore, I recommend its publication.

Response to Reviewer #1

Report on the resubmitted paper 481537_1 “Gradient and curl optical torques” by Xiaohao Xu et al.

Upon resubmission, the authors have carefully addressed the previous remarks, and now the paper is essentially improved. It is only one minor point relating the momentum-curl torque (3 rd Eq. (3)), that I overlooked in the 1st review. Actually, in some cases the term $\propto \nabla \times \mathbf{p}$, \mathbf{p} being the Poynting vector or a certain its constituent, represents a sort of the “true” electromagnetic spin (see, for example, [47], Transverse spin and the hidden vorticity of propagating light fields <https://doi.org/10.1364/JOSAA.466360>; Transverse spinning of unpolarized light <https://doi.org/10.1038/s41566-020-00733-3>). In such cases, the “momentum-curl” contribution is, at least partially, “captured” by the “pure-spin” torque (1). Accordingly, the “momentum-curl” torque can be “felt” even by dipole particles. Am I right? If so, this point should be briefly commented to avoid confusing ambiguities.

Response (R1.1): Thank you for your careful reading of the revision and your question. As regards the connection between $\nabla \times \mathbf{p}$ and the spin, recent progress in field theory has established the momentum curl as the ‘Poynting’ part of the spin (only in dual-symmetric form) [45,46]. These two quantities can even be proportional to each other for some fields (e.g., evanescent waves with purely transverse spin), as indicated by Ref. [47] and [<https://doi.org/10.1364/JOSAA.466360>] the reviewer mentioned. This point is commented and the JOSAA paper is quoted as Ref. [48] in the revision. However, it might be potentially contentious to say that the curl torque can thus be felt by dipole particles, even though the dipolar pure-spin torque can really be written in terms of momentum curl for these special fields. After all, to derive this “curl torque”, one needs to start from the pure-spin torque, Eq. (1), and then recast its mathematical form, during which process no physics of light-matter interactions will be involved. We believe the view on this issue varies from person to person.

As to the other materials, I can only suggest some small technical corrections:

P. 3, col. 1, line 3 from the bottom: “ $\nabla \times \mathbf{P}$ ”  “ $\nabla \times \mathbf{p}$ ”?

R1.2: Corrected.

P.3, col. 2, beneath Eq. (6): “four-plane waves”  “four plane waves”?

R1.3: Corrected.

P. 5, col. 2, line 10: “Lorenz-Mie method-based results”  “results based on the Lorenz-Mie method”?

R1.4: Refined.

P. 6, col. 1: “washboard potential is slightly titled”  “washboard potential is slightly

tilted”?

R1.5: Corrected.

P. 6, col. 2: “Richard-Wolf method” “Richards-Wolf method”?

R1.6: Corrected.

P. 8, col. 2: “Due to symmetry, $Tz(\alpha)$ on an arbitrary particle must be a periodic function of α ”  $Tz(\alpha)$ must be a periodic function of α because α and $\alpha + 2\pi$ describe the same physical configuration; this does not depend on symmetry.

R1.7: Revised.

P. 11: Ref. 62 is incomplete.

R1.8: The information about this reference is complemented.

I think, the suggested corrections are technical and can be made during the final text polishing. I support the paper publication.

R1.9: We greatly appreciate your suggested corrections, which has been made in the final polish.

Response to Reviewer #2

The manuscript by Xu et al. introduces a comprehensive framework for describing the torque exerted on a Mie particle by any electromagnetic field. The study reveals that beyond dipolar approximation, torques originating from helicity gradients and momentum-curl of light emerge. Moreover, it explores situations where structured light beams induce negative optical torque (NOT) and lateral optical torque (LOT). The authors responded extensively to my queries. My concern regarding the paper's incremental nature compared to prior work by one of the authors (Nieto-Vesperinas, PRA 92, 043843 (2015)) remains valid. However, the manuscript's inclusion of discussions on LOT affecting anisotropic particles significantly enhances its quality and broadens its appeal to the scientific community. Therefore, I recommend its publication.

Response: We appreciate your recommendation and positive comments on our revisions. Still, we would re-emphasize that the gradient and curl components do not exist in the dipole approximation Prof. Nieto-Vesperinas dealt with throughout his PRA paper, and our theoretical framework cannot be derived from the results therein.